# Urban Shrinkage and Urban Vitality Correlation Research in the Three Northeastern Provinces of China

**DOI:** 10.3390/ijerph191710650

**Published:** 2022-08-26

**Authors:** Yihao Jiang, Zhaojin Chen, Pingjun Sun

**Affiliations:** School of Geographical Sciences, Southwest University, Chongqing 400715, China

**Keywords:** urban shrinkage, urban vitality, shrinking cities, two-step diagnostic method, correlation analysis, the three northeastern provinces of China

## Abstract

In the global trend of urban shrinkage, urban vitality, as one of the important representations of high-quality urban development, has become a breakthrough. More and more scholars advocate to awaken urban vitality, so as to realize the high-quality development of shrinking cities. This paper takes the municipal districts of 34 cities in the three northeastern provinces of China as study areas, based on the broad concept of urban shrinkage, selects the indicators of population, economy and society, and uses the “two-step diagnostic method” which is consistent with Chinese conditions to identify the urban shrinkage from 2010 to 2018. In this research, the indexes of economic, social, cultural, environmental and spatial dimensions are selected, and the urban vitality and the vitality of each dimension from 2010 to 2018 are calculated and analyzed by using the entropy weight method (EWM). Then, this paper analyzes the correlation between urban shrinkage and urban vitality by Pearson correlation coefficient. The results show that: (1) urban shrinkage in the three northeastern provinces of China has become a regional remarkable phenomenon, which is also an inevitable process in some regions of China and even the world; (2) overall, the urban vitality of cities in the three northeastern provinces of China is steady and rising a little, and there is an obvious spatial agglomeration pattern like “central city polarization”; (3) there is a significant correlation between urban shrinkage and urban vitality, that is, the lower the degree of urban shrinkage, the higher the urban vitality, showing the opposite trend in the process of urban development; (4) the influence of urban shrinkage on each dimension of urban vitality is different, and the correlation results are different, too. In the planning process of shrinking cities in the future, paying attention to the relationship between urban vitality and urban shrinkage, conducting benign guidance on this basis, and adjusting urban vitality elements of different dimensions to stimulate urban development power can enhance urban competitiveness and achieve better development.

## 1. Introduction

At present, the whole world, especially Asia, is experiencing rapid urbanization. The rapid urbanization has brought about the obvious expansion of the city, which seems to indicate that the city will grow for a long time or even infinitely under the tide of population increase, economic growth and area expansion [1]. However, the phenomenon of urban shrinkage since the second half of the 20th century has broken this inference. UN-HABITAT points out that during the period of 1970–1990 and 1990–2014, the ratio of cities which experienced population loss remained at around 10% globally. Relevant research also finds that the phenomenon of population loss appeared in the process of urban development of developed countries such as Germany, Japan, and the United States and developing countries represented by China [2]. People have to admit that urban shrinkage has gradually evolved into a global problem, and the phenomenon of its population reduction has brought severe challenges to the comprehensive competitiveness of cities, which has become an urgent problem to be solved in the fields of urban planning and regional research. Under this background, the urban vitality, which is one of the important representations of the high-quality development of the city, has become a new breakthrough idea, gradually appearing in the context of urban development based on pervasive shrinkage [3]. As one of the actual expressions of the exuberant vitality of urban space, urban vitality reflects the ability and potential of cities’ development to a certain extent and becomes a new source of urban competitiveness. Nowadays, under the realistic predicament that the phenomenon of urban shrinkage continues to appear all over the world, more and more scholars advocate awakening the vitality of cities and realizing the goals of the high-quality development of cities by adjusting the vital elements of cities and improving cities’ comprehensive competitiveness [4]. The phenomenon of population reduction and negative economic growth caused by urban shrinkage seems to have become an important obstacle to urban development, and the promotion of urban vitality is a breakthrough to solving this problem naturally. Therefore, it is time for academia to further sort out the relationship between urban shrinkage and urban vitality, find out the theoretical and logical connection between urban shrinkage and urban vitality, and then better guide cities to enhance urban vitality in the process of shrinkage in order to help cities obtain stronger development power and complete the transformation from general urbanization to high-quality development.

Since the German scholar, Häußermann put forward the concept of urban shrinkage in 1980s, urban shrinkage has always been one of the hot words in urban research [5]. At present, scholars from all over the world study urban shrinkage by mainly focusing on connotation interpretation, type identification, planning response, etc. (1) For connotation interpretation, current scholars believe that there are two definitions of urban shrinkage including a narrow and broad definition. The narrow sense of urban shrinkage is mainly characterized by the reduction of absolute population. Shrinking City Research Network defines urban shrinkage as a phenomenon of a city with at least 10,000 people, with population loss of more than two years, structural crisis and economic transformation [6]. Shrinking City Project holds that urban shrinkage should meet the requirements of population loss of more than 10% and more than 1% population loss every year [6,7]. However, with the deepening of research, some scholars suggest that cities are spatial entities formed by the interactions of population, economy and space, and the reduction of population cannot fully reflect all the characteristics in the process of urban shrinkage, which is accompanied by the adjustment of economic structure and the change of social environment. Therefore, they believe that urban shrinkage in a broad sense should take into account the changes in population, economy, space and environment, etc. [2,8,9,10,11]. It is worth mentioning that, regardless of the definitions, scholars have reached a consensus that the process of urban shrinkage is accompanied by the loss or attenuation of urban development elements. Under the background of the scarcity of urban development resources, its comprehensive competitiveness will inevitably be affected by this, resulting in the decline of advantages of location and the weakness of urban development. (2) For the identification of urban shrinkage types, the current scholars mainly use three identification types: population change rate division, spatial form division and shrinkage cause division. According to the change rate of shrinking urban population, Chen et al. [12] divided the shrinkage types into isolated shrinkage, continuous shrinkage and staggered shrinkage by studying the urban shrinkage in the Yellow River basin from the perspective of population. According to the population and economic data of Liu et al. [13], the shrinkage types are further divided into absolute and relative shrinkage. According to the shrinking urban space form, it can be divided into “perforated”, “doughnut” and “peripheral” types. According to the relationship between shrinking cities and surrounding cities, it can be defined as marginal core dependent shrinkage [14]. (3) For planning response, scholars mainly try to solve the development problems of shrinking cities with ideas such as “resilient city”, “smart shrinkage” and “inclusive city” [15,16], trying to improve urban vitality from several aspects in order to solve the development problems of shrinking cities.

The concept of urban vitality originated from Jacobs’ description of active street life created by a large number of pedestrians [17]. Since then, scholars have never reached a unified understanding of urban vitality: Maas holds that successfully maintaining a vibrant area depends on street life and its social interaction; Landry thinks that economic, social and physical environmental factors interact to form vitality [17,18]. At the same time, the existing related study areas are mostly concentrated on streets and neighbors. Although different scholars pay different degrees of attention to urban vitality, which leads to various expressions, it has become a consensus that urban vitality has many attributes such as economy, society, culture and multiple influencing factors. Therefore, different from the treatment of street vitality as urban vitality based on Jacobs’ theory, nowadays, there are many definitions focusing on the macro meaning of urban vitality in the academia. From the perspective of urban sociology, urban vitality is regarded as a concrete representation of urban competitiveness, which is subdivided into three parts: economic vitality, social vitality and cultural vitality, showing the ability of cities to provide facilities and space for urban residents’ activities [19]. At the same time, as for the measurement methods of urban vitality, plenty of scholars measures urban vitality on the street scale based on global positioning system, point of interest (POI), mobile phones, integrated circuit cards, taxis and other data and there are also some scholars who measures the urban vitality in a macro sense according to urban statistical data, bulletins and yearbooks [20]. It is worth noting that more and more scholars have applied urban vitality to other fields, such as sociology, criminology and public health, etc., and have started a critical reflection on the past urban expansion through the possible impact of urban shrinkage on urban vitality [21,22,23].

Generally, although there have been many related studies on urban shrinkage and urban vitality, their connotation, measurement, cause identifications and response of plannings have all been discussed, little research has been done on the correlation between them, especially the urban shrinkage and urban vitality correlation researches, which should be an innovative direction. However, from the existing unilateral researches on urban shrinkage or urban vitality, we can find the possible internal relationship between them: Firstly, because each city is a spatial complex with multiple factors, the influencing factors related to its development are often diverse, so the evaluation system constructed by the related researches on urban shrinkage and urban vitality often has a strong correlation, which is reflected in economic, social and environmental indicators; Secondly, the changes of urban development factors in the process of urban shrinkage often leads to the decline of urban competitiveness, and urban vitality is regarded as one of the specific manifestations of urban competitiveness. Therefore, the process of urban shrinkage often leads to the corresponding fluctuations of urban vitality, which is finally reflected in the comprehensive level of urban competitiveness. Thirdly, the decision makers can boost the urban vitality in the reverse direction through the concepts of smart shrinkage and resilient city, and they also put forward the advocacy of “plans of shrinkage” accordingly [16,23,24,25]. On this basis, this paper holds that there is a certain logical connection and mechanism between urban shrinkage and urban vitality. In a word, urban shrinkage and urban vitality show a certain degree of correlation because they jointly influence urban development and eventually settle down in urban competitiveness. However, considering the diversity of influencing factors, the relationship between urban shrinkage and urban vitality cannot be simply understood as a simple linear relationship with the same increase and decrease, or a trade-off between the increase and the decrease. Urban shrinkage and urban vitality form their own systems, and each system has constantly changing elements and subsystems corresponding to different indicators. The systems are connected by indicators with common or similar points, which then interact with each other. The results of these interactions can often reflect the specific relationship between the two systems, which shows the correlation between urban shrinkage and urban vitality (Figure 1). Furthermore, the influencing factors of urban vitality are diverse, and the degree of correlation between urban shrinkage and urban vitality will also be differentiated due to different dimensions of urban vitality. Based on the thinking and speculation, this research starts with urban shrinkage and urban vitality, respectively, identifies and measures each of them, and then analyzes and explores their relationship with the method of correlation analysis. Exploring the relationship between urban shrinkage and urban vitality in the aspects of economy, society, culture, etc. has a considerable effect on the deep excavation of urban vitality of shrinking cities, and the results will directly guide the urban development planning of cities experiencing shrinking and have an enlightening effect on the research direction of shrinking-cities planning [26,27]. Therefore, the research on the correlation between urban shrinkage and urban vitality should be put on the research agenda as soon as possible to solve the existing theoretical dilemma and practical problems.

## 2. Materials and Methods

### 2.1. Study Areas

As the case areas for the study, this paper selects the three northeastern provinces of China, namely Heilongjiang Province, Jilin Province and Liaoning Province, with a total area of 787,300 km^2^, including 34 prefecture-level municipalities, one prefecture and one autonomous prefecture (Figure 2). Northeast China is one of China’s four major economic sectors, a relatively complete and independent economic zone, and also an old industrial base and resource-based city-intensive area [28]. The three northeastern provinces of China are adjacent to many countries, which is an important window for China to open to Northeast Asia. From this point of view, the three northeastern provinces of China are of irreplaceable importance to China and neighboring countries. At the same time, due to the external factors such as globalization, transformation of urban science and technology innovation, and internal factors such as over-exploitation of energy and imbalance of economic structure, the three northeastern provinces of China gradually lost their original location advantages in the development process in recent years, and their regional competitiveness declined obviously, resulting in large-scale urban shrinkage. At the same time, the Chinese government has always paid attention to the urban vitality of the three northeastern provinces of China and put forward “the Revitalization of the Northeast Strategy” [29], in order to improve their urban vitality, realize the comprehensive promotion of urban competitiveness, and reach the high-quality development of shrinking cities. Therefore, under the comprehensive effect of the remarkable shrinkage phenomenon and the measures of government policies to guide the revitalization of urban vitality, the three northeastern provinces of China have undoubtedly become one of the best areas to explore the relationship between urban shrinkage and urban vitality in the context of Chinese developments. This paper is intended to explore the problems of cities, so the focus area is cities without villages and small towns, specifically, the municipal districts of 34 prefecture-level municipalities in the three northeastern provinces of China.

### 2.2. Data Sources

The data of this research is from 2010 to 2018, and the main sources are the China City Statistical Yearbook (https://data.cnki.net/yearbook/Single/N2022040095, accessed on 19 August 2022), China Urban Construction Statistical Yearbook (https://data.cnki.net/yearbook/Single/N2021110010, accessed on 19 August 2022), China Statistical Yearbook for Regional Economy (https://data.cnki.net/yearbook/Single/N2015070200, accessed on 19 August 2022), Heilongjiang Statistical Yearbook (https://data.cnki.net/yearbook/Single/N2022030241, accessed on 19 August 2022), Jilin Statistical Yearbook (https://data.cnki.net/yearbook/Single/N2022010252, accessed on 19 August 2022) and Liaoning Statistical Yearbook (https://data.cnki.net/yearbook/Single/N2022010298, accessed on 19 August 2022) from 2011 to 2019. In this study, there are two ways to deal with the missing data in the above statistical yearbooks. In principle, it is supplemented according to the official websites of relevant departments of various provinces and cities and the statistical bulletins of national economic and social development. If the data are still missing, this research interpolates the data by taking the average of the years before and after. The data of this period is selected because the problems of urban development in the three northeastern provinces of China have been obviously exposed during the last decade [30]. At the same time, considering the uncertainty brought by Corona Virus Disease 2019 (COVID-19) epidemic on all aspects of urban development and the influence of COVID-19 epidemic on statistical work, in order to avoid large fluctuations of statistical data during the epidemic, this research does not include the data in 2019 and after. Because the data of Da Hinggan Ling Prefecture in Heilongjiang Province and Yanbian Chaoxian (Korean) Nationality Autonomous Prefecture in Jilin Province are missing in large numbers and the urban system is special, this research eliminates them.

### 2.3. Methods

#### 2.3.1. Identification of Urban Shrinkage

Urban shrinkage is a series of reactions based on the loss of urban population. In the later period, there will be more changes in economy, society and resources around the changing population [29]. In previous research, scholars have gradually realized the limitations of taking population loss as the only identification basis of urban shrinkage. In the process of urbanization in developing countries, urban population mobility is a normal phenomenon and an inevitable process, which may not cause significant changes in urban functions and quality. The process of urbanization in China is also inseparable from a large-scale population change. Only when the population decreases continuously and on a large scale, which leads to the retrogression of urban economic and social development, can we think that the city has shrunk [31]. According to such characteristics, this research constructs a comprehensive identification system of shrinking cities with population as the core index, supplemented by economic and social indicators which form a “population-economy+society” system and selects the annual average population of municipal districts, gross domestic product (GDP), per capita disposable income, total retail sales of social consumer goods, the proportion of the output value of the tertiary industry to GDP, and changes in the number of doctors in municipal districts to characterize the phenomenon of urban shrinkage (Table 1) [28]. Among them, population change is the most basic and direct embodiment of urban shrinkage, but it is not the only basis. GDP, per capita disposable income, and industrial structure changes can reflect the changing trend of shrinking cities in the economic dimension from a macro perspective, which is a comprehensive embodiment of urban economic conditions [28]. The change of the number of doctors and total retail sales of social consumer goods can be used as a representative index of social dimension to show the deep-seated problems of shrinking cities, reflecting the social security and service capacity of cities [32].

According to the general process of urban shrinkage and the characteristics of urbanization in China, referring to the “two-step diagnostic method” proposed in 2021 by Sun et al. [28], population shrinkage is the primary characteristic of urban shrinkage, not the only evidence, and it has been generally recognized by the academia. This study also takes the change of population as the first criterion to identify shrinking cities, and then further judges the specific situation of urban shrinkage with other indicators. Therefore, this research holds that the progressive method is most suitable for identifying urban shrinkage. Referring to this method, this research measures the annual average population loss of municipal districts in a continuous period of time first, and referring to the existing research results and China’s policies, this research preliminarily determines the cities with negative population growth rate ($X_{1}$) for three consecutive years as shrinking cities, that is, there is no shrinking city phenomenon in the cities that failed the first judgment; Then, on this basis, the shrinkage of other indicators is measured. Similarly, in order to eliminate the influence of normal fluctuations, this research determines the index with negative growth rate for three consecutive years as shrinkage state and calculated its change rate between the beginning and ending years of shrinkage ($X_{i}$), and then calculates the average change rate of all indicators with shrinkage, and got the secondary score value (A). If a city that has passed the first judgment does not have any shrinking indicators, it will not pass the second judgment. If a city shrinks twice in a certain judgment from 2010 to 2018, the final population growth rate ($X_{1}$) or secondary score value (A) is the average of the two shrinking processes. Next, according to the population growth rate ($X_{1}$) and the secondary score (A), this research divides urban shrinkage into four stages: slight shrinkage, preliminary shrinkage, middling shrinkage and severe shrinkage.

The population growth rate ($X_{1}$), the change rate between the beginning and ending years of shrinkage ($X_{i}$) index and the secondary score value (A) are calculated as follows:(1)$$X_{1}=\frac{X_{t}-X_{0}}{X_{0}}$$
(2)$$X_{i}=\frac{X_{it}-X_{i0}}{X_{i0}}$$
(3)$$A=\frac{\Sigma X_{i}}{5}\left(i=2,\;3,\;4,\;5,\;6\right)$$
where $X_{t}$, $X_{0}$ represent the values of the population in the shrinkage start year, the population in the shrinkage end year, respectively, and $X_{it}$, $X_{i0}$ represent the value of the *i*-th index in the start year of shrinkage and the value of the *i*-th index in the end year of shrinkage.

After the judgments, this study needs to determine the weight of the population growth rate ($X_{1}$) and the secondary score value (A), and then take the weighted average of the two as the urban shrinkage score (T). Referring to the methods of many research results and combining with the actual situation of the study areas, this paper posits that the population reduction has a great influence on the subsequent shrinkage process, so the proportion of cities that have passed the second judgment in the number of cities that have passed the first judgment (α) and the proportion of cities that have failed the second judgment in the number of cities that have passed the first judgment (1−α) are used as alternative weights [33,34,35]. Because population change is the primary sign of urban shrinkage, this study gives the higher one to $X_{1}$ [33]. According to the urban shrinkage score (T), the basis for delineating the urban shrinkage stage is shown below (Table 2).

The urban shrinkage score (T) is calculated as follows:(4)$$\mathrm{If}\;\alpha \geq 0.5,\;\mathrm{then}\;T=\alpha *X_{1}+\left(1-\alpha \right)*A$$
(5)$$\mathrm{If}\;\alpha <0.5,\;\mathrm{then}\;T=\left(1-\alpha \right)*X_{1}+\alpha *A$$

#### 2.3.2. Measuring of Urban Vitality

In all stages of urban development, urban vitality is the basic element to realize urban quality [36]. As one of the representations of comprehensive competitiveness of cities, macroscopical urban vitality mainly reflects the level of urban economic and social development, as well as the resource base and progress potential of urban culture, environment and space, etc. [23,24]. Urban vitality reflects different human activities and the interactions between human beings and cities [37,38,39]. All indicators of urban vitality reflect the ability of the city at different times and corresponding aspects, and together, they can explain the flow of elements in the region where the city is located.

Combining with the existing theories and the comprehension above, this paper holds that the selection of indicators of urban vitality should be carried out from five aspects: economy, society, culture, environment and space, and these aspects should be reasonably explained by a number of secondary indicators [18,40]. Economic vitality is an intuitive expression of urban strength, an important embodiment of urban development quality, and the best reflection of urban development potential. Social vitality closely revolves around human activities in the cities, which can reflect the driving force of the city with the most active elements in the cities. Cultural vitality is related to the historical background, spiritual character and cultural landscape of the city, and has a certain influence on the path of urban development [41]. Environmental vitality shows the carrying capacity of cities through the level of vegetation cover, ecological consumption, public health and other factors in the cities. Spatial vitality consists of the level of public facilities, public services, land use and infrastructure in cities. Under the background of possible urban shrinkage in the three northeastern provinces of China, macro-scale urban vitality can be used as a comprehensive standard to judge the absolute strength and relative level of cities in a certain region.

Finally, based on the existing research results and the principles of scientificity, systematicness and effectiveness, this paper selects 20 s-level indicators belonging to 5 dimensions to construct the evaluation index system of urban vitality (Table 3).

In this research, EWM based on standardization is used to measure the urban vitality of the three northeastern provinces of China. The specific steps are as follows.

(1) Standardize the positive indicators (the bigger the better) and negative indicators (the smaller the better) of urban vitality from 2010 to 2018:

① Positive indicators:(6)$$X_{ij}^{*}=\frac{X_{ij}-X_{ij,min}}{X_{ij,max}-X_{ij,min}}\;$$

② Negative indicators:(7)$$X_{ij}^{*}=\frac{X_{ij,max}-X_{ij}}{X_{ij,max}-X_{ij,min}}\;$$
where $X_{ij}^{*}$ is the value of standardized data sample, $X_{ij}$ is the sample value of the *i*-th city in the *j*-th index (*i* = 1, 2, 3, …, 34; *j* = 1, 2, 3, …, 20), $X_{ij,max}$ is the maximum value of the *i*-th city in the *j*-th index, and $X_{ij,min}$ is the minimum value of the *i*-th city in the *j*-th index.

(2) Use EMW to determine the entropy and weight of each index:

① The proportion of the sample value of the *i*-th city in the *j*-th index each year ($y_{ij}$):(8)$$y_{ij}=\frac{x_{ij}^{*}}{\sum_{j=1}^{34}X_{ij}^{*}}$$

② Information entropy of the *j*-th index each year ($e_{j}$):(9)$$e_{j}=-K*\sum_{j=1}^{34}y_{ij}\ln y_{ij},\;\mathrm{where}\;K\;\mathrm{is}\;a\;\mathrm{constant},\;\mathrm{and}\;K=\frac{1}{\ln34}$$

③ The weight of the *j*-th index each year ($w_{j}$):(10)$$w_{j}=\frac{1-e_{j}}{\sum_{j}\left(1-e_{j}\right)}$$

(3) Obtain the comprehensive scores and rankings of cities each year:

① Use weighted sum to calculate the comprehensive score (S) of each city every year, and use it as the value of urban vitality:(11)$$S=\overset{34}{\underset{j=1}{\sum }}w_{j}*y_{ij}\;$$

② Rank according to the comprehensive scores from big to small.

Repeating the above process with the data from 2010 to 2018, we can obtain the annual urban vitality values and ranking changes of cities in the three northeastern provinces of China. In addition, according to the same method, this study calculates the comprehensive scores of five dimensions, namely, economic vitality, social vitality, cultural vitality, environmental vitality and spatial vitality, for the follow-up analysis.

#### 2.3.3. Correlation Analysis between Urban Shrinkage and Urban Vitality

According to the characteristics and applicable conditions of the data, this research uses Pearson correlation coefficient to analyze the correlation between the comprehensive score of urban vitality and the score of urban shrinkage from 2010 to 2018.

The correlation coefficient (r) is calculated as follows:(12)$$r=\frac{\sum_{i=1}^{34}\left(S_{i}-\bar{S}\right)\left(T_{i}-\bar{T}\right)}{\sqrt{\sum_{i=1}^{34}\left(S_{i}-\bar{S}\right)2}\sqrt{\sum_{i=1}^{34}\left(T_{i}-\bar{T}\right)2}}$$
where $S_{i}$, $\bar{S}$, $T_{i}$ and $\bar{T}$, respectively, represent the comprehensive score of urban vitality of the *i*-th city, the average of comprehensive scores of all urban vitality, the urban shrinkage score of the *i*-th city and the average of all urban shrinkage scores.

After obtaining the correlation coefficient value, the significance of Pearson correlation coefficient is tested. In this research, Pearson correlation coefficient is used to construct a statistic (p):(13)$$p=r*\sqrt{\frac{34-2}{1-r2}}$$

In this test, the original hypothesis is H0: r = 0, that is, there is no linear correlation between comprehensive scores of urban vitality and urban shrinkage scores, and the alternative hypothesis is H1: r ≠ 0, that is, there is a linear correlation between comprehensive scores of urban vitality and urban shrinkage scores. When p<0.01, this paper rejects the original hypothesis and accepted the alternative hypothesis at 99% confidence level. When p<0.05, this paper rejects the original hypothesis and accepted the alternative hypothesis at 95% confidence level.

## 3. Results

### 3.1. Basic Situations of Urban Shrinkage in the Three Northeastern Provinces of China

After the identification by “two-step diagnostic method”, this research obtains the basic situation of urban shrinkage in the three northeastern provinces of China from 2010 to 2018. As shown in Table 4, the urban shrinkage in the three northeastern provinces of China has following characteristics: (1) From 2010 to 2018, 21 cities have shrunk in different degrees, which reflects the general trend of urban development in the three northeastern provinces of China. According to the identification by “two-step diagnostic method”, except Harbin, Daqing, Mudanjiang, Heihe, Changchun, Liaoyuan, Songyuan, Shenyang, Dalian, Jinzhou, Yingkou, Panjin and Chaoyang, the other cities have experienced urban shrinkage of different scales, and the number of these cities is more than that announced by the Ministry of Housing and Urban-Rural Development of China in April 2019. (2) Among the 21 shrinking cities, 15 cities have passed the second judgment, that is, there are other shrinking phenomena except population shrinking. From this, it can be preliminarily judged that urban population shrinking is likely to trigger the subsequent shrinking process, resulting in regional urban shrinking phenomenon. (3) According to the identification results, there are nine cities in the middling shrinkage stage in the three northeastern provinces of China. Middling shrinkage generally shows that the population is greatly reduced or slightly reduced, while other indicators are obviously reduced. (4) Generally speaking, the phenomenon of urban shrinkage in the three northeastern provinces of China is not a single case, but it has shown certain regional commonalities and inter-city differences. Urban shrinkage is widely distributed in the three northeastern provinces of China, accounting for 61.76% of the total number of cities. At the present stage, urban shrinkage is an inevitable process of urbanization, and its relationship and function with urban vitality are worth exploring, with a view to providing ideas for urban development in the three northeastern provinces of China and even other regions [4].

### 3.2. Spatial Characteristics of Urban Vitality in the Three Northeastern Provinces of China

Table 5 shows the changes of urban vitality values and rankings of cities in the three northeastern provinces of China from 2010 to 2018. In order to intuitively reflect the spatial pattern and temporal evolution of urban vitality in the three northeastern provinces of China, this research uses ArcGIS to display the annual urban vitality values of each city on the maps (Figure 3). When mapping, this research selects the natural breakpoints of the comprehensive scores of the middle year (2014) as the natural breakpoints of all years for grading display, which can show the absolute size and relative level of urban vitality at the same time and can more scientifically show the temporal and spatial change process of urban vitality.

It can be seen from Figure 2: (1) In the nine years, the urban vitality in the three northeastern provinces of China shows an overall upward trend, while the values of most cities fluctuated, not rising and falling steadily. (2) The high values of urban vitality in the three northeastern provinces of China show a trend of gathering towards centers, that is, the three provincial capitals and Dalian become the center of the region, respectively, which promotes the value of surrounding cities to rise and forms a large-scale and regional linkage when their own urban vitality is at a high level for a long time. Previously, a large number of studies have shown that in Asia, places with high urban vitality are often located in central cities [37]. This research reveals that in the three northeastern provinces of China, the places with high urban vitality are all in the cities with strong central characteristics, such as Harbin, Changchun, Shenyang and Dalian. (3) The urban vitality of the three northeastern provinces of China, especially the Liaodong Peninsula city group, is generally rising but polarized towards the middle. (4) The vitality of cities in northern Heilongjiang, southern Jilin and western Liaoning has been sluggish for a long time, changing little, and it has not been led by other cities to develop. Based on the existing theories and the above analysis, this paper speculates that there is a certain correlation between urban shrinkage and urban vitality in the three northeastern provinces of China, which is worthy of further exploration.

### 3.3. The Relationship between Urban Shrinkage and Urban Vitality

According to the method and process of correlation analysis, this research finds that there is a significant positive correlation between the comprehensive scores of urban vitality and the scores of urban shrinkage in the three northeastern provinces of China from 2010 to 2018. However, it is worth noting that the urban shrinkage scores used in the research are negative, and the positive correlation only aims at the change between the two scores themselves and does not reflect the relationship between urban shrinkage and urban vitality. Specifically, the more serious the urban shrinkage, the lower its urban vitality. From 2010 to 2018, the correlation coefficient values are 0.386, 0.451, 0.438, 0.469, 0.488, 0.457, 0.463, 0.437 and 0.457, respectively, all of which pass the significance test. The result shows that the degree of urban shrinkage of cities in the three northeastern provinces of China will have a positive impact on the relative size of urban vitality, and the impact reaches its peak in the middle year (2014).

According to the index systems of urban shrinkage and urban vitality, the relationship between urban shrinkage and urban vitality is understandable. In a certain year, the more obvious the shrinkage phenomenon is, the lower its urban vitality is. It has been proved that there is a strong correlation between urban shrinkage and urban vitality in time and space. The correlation coefficient value between urban vitality and the urban shrinkage reaches the highest value in 2014, indicating that the change of urban vitality in 2014 is in the middle stage, which can represent the average level of urban vitality in nine years to a certain extent. The correlation coefficient value between the comprehensive urban vitality score and the urban shrinkage score is the lowest in 2010, because 2010 is the starting year of this research, so the trend of urban shrinkage of that year cannot be judged.

To sum up, urban shrinkage is the result of a long period, while urban vitality is the value that changes yearly. They represent the stage state and real-time capability of cities, respectively. The correlation shows that different degrees of urban shrinkage have different effects on urban vitality. At present, for the whole region, urban shrinkage has shaped the spatial pattern of urban vitality. For some cities, urban shrinkage slows down the promotion of urban vitality, but it does not completely limit the exertion of urban vitality.

However, from 2010 to 2018, although urban shrinkage and urban vitality are significantly correlated, the correlation coefficient values are not more than 0.5, that is, they are not completely correlated. It suggests that the role between urban shrinkage and urban vitality is actually complex and diverse, which needs further exploration and analysis.

### 3.4. The Relationship between Urban Shrinkage and Various Dimensions of Urban Vitality

In order to explore the specific impact of urban shrinkage on urban vitality, this research conducts correlation analysis between the comprehensive scores of economic vitality, social vitality, cultural vitality, environmental vitality and spatial vitality of cities in the three northeastern provinces of China from 2010 to 2018 and the urban shrinkage scores (Table 6) and compares the five groups of results.

The correlation coefficient values of comprehensive economic vitality scores and urban shrinkage scores are 0.363, 0.401, 0.386, 0.402, 0.411, 0.398, 0.426, 0.408, 0.411, respectively, all of which show 0.05 level significance. The results show that there is a significant positive correlation between urban economic vitality and urban shrinkage, and the change trend of correlation coefficient value is similar to that obtained by the above analysis. This result is not unexpected. According to the method of this research, the judgment of urban shrinkage cannot be separated from economic factors. The economy of the cities in the shrinkage stage often does not perform well, such as the sharp drop of foreign investment, the imbalance of industrial structure, and the shrinking of market scale. The shrinkage of various economic indicators in cities naturally leads to the reduction of economic vitality.

The correlation coefficient values of comprehensive scores of urban social vitality and urban shrinkage scores are generally low, and only the data of 2016 pass the significance test. It can be shown that, the correlation between urban social vitality and urban shrinkage is weak. This is because the mechanism of urban shrinkage on social vitality is different from that on economic vitality. Although social vitality is related to population, urban shrinkage is a comprehensive and phased process, which cannot be completely summarized by population changes. Moreover, the indicators of social vitality have been stripped of dimensions, often reflecting the average level of every 10,000 people, so the change rate of data is buffered. At the same time, the social vitality base of each city is quite different, so it is not affected by urban shrinkage obviously.

The correlation coefficient values between the comprehensive score of urban cultural vitality and the score of urban shrinkage show a trend of first increasing and then decreasing on the whole, with the data in 2012 reaching the peak, and the data in 2017 and 2018 failing the significance test. There is a certain correlation between the cultural vitality and the urban shrinkage, but this relationship gradually weakens after reaching the peak. On the one hand, culture belongs to people’s high-level needs, and it also represents the historical origin and spiritual wealth of cities. Urban shrinkage brings new challenges to urban population and economic and social development, which will have an impact on the vitality of urban culture, and therefore there is a certain correlation between the two. On the other hand, in recent years, China has taken the road of cultural self-confidence, pays more and more attention to the excavation and construction of urban culture, and actively promotes the upgrading of cultural vitality. Therefore, the impact of urban shrinkage on cultural vitality gradually decreases.

The correlation coefficient values of urban environmental vitality comprehensive score and urban shrinkage score fluctuates greatly in nine years, among which the data of 2010, 2012 and 2013 fails the significance test, and the correlation coefficient values of other years fluctuates obviously, not displaying specific laws. Therefore, there is a certain correlation between urban environmental vitality and urban shrinkage, but the degree of correlation between them is not clear and changes at any time. The vitality of the environment mainly reflects the ecological value, sanitary quality and supply capacity of the city. According to China’s national conditions, the first two indicators generally increase year by year and will not decrease with the shrinkage of the city. Urban shrinkage affects the overall level of social production through a series of changes in population, economy and society, coupled with the special climate in the three northeastern provinces of China, which in turn can change the supply capacity of cities to a certain extent, such as causing a temporary regional shortage of coal and electricity. However, this process only happens by chance and there is uncertainty. Therefore, there is no obvious law in the impact of urban shrinkage on environmental vitality.

The correlation coefficient values of comprehensive urban vitality score and urban shrinkage score are all close to 0, and the value of p is far beyond the reasonable range. It shows that there is no correlation between urban spatial vitality and urban shrinkage. According to the index system of this research, the spatial vitality corresponds to the infrastructure construction and support capacity of cities. In developing countries, especially China, infrastructure construction and support capacity are constantly improving. Generally, the index corresponding to urban spatial vitality is likely to improve continuously, so it will hardly be affected by urban shrinkage.

## 4. Discussion

### 4.1. Urban Shrinkage has Become a Remarkable Regional Phenomenon in the Three Northeastern Provinces of China

China’s reform and opening up has enabled many regions and cities to achieve rapid economic and social development in the last few decades and has also accelerated the urbanization process of the whole country. As the largest developing country in the world, China’s rapid urbanization process is accompanied by drastic changes in population, economy, society, land and resources [46,47]. However, as one of the important economic sectors, the three northeastern provinces of China have to face a series of development difficulties, and the phenomenon of urban shrinkage has become very significant [28]. In the series of processes, due to the mobility of development factors, the outflow of population in some cities has enabled the resources in the region to be optimally allocated, the urban functions to be optimized, the social structure to be improved, and the economy to develop well; However, the population changes in some cities have the opposite effect, triggering a chain of reactions, resulting in substantial urban shrinkage. Therefore, the current urban shrinkage in China is a systematic and progressive process, which should be identified by comprehensive index system and measurement methods. Generally speaking, this paper posits that “two-step diagnostic method” is suitable for this step.

Through the identification of “two-step diagnostic method”, this research finds that the urban shrinkage in the three northeastern provinces of China is a collective, continuous and regional phenomenon, which is not only the inevitable result of the transformation of urban functional structure after urbanization has reached a certain level, but also the concentrated reflection of many aspects of urban development problems. At present, there is no unified standard for the definition and identification of urban shrinkage by governments and scholars all over the world, and there are also mixed opinions on the phenomenon of urban shrinkage itself and its possible results [2,6,28]. The attitude of some papers towards urban shrinkage is neutral, that is, urban shrinkage is not equal to urban recession, and it can be reasonably intervened and guided to create new development models. However, according to the characteristics of identification methods and index systems, this paper inevitably holds a few negative views on urban shrinkage. This theoretical dilemma is the commonality of many research results, which needs to be broken. Since the beginning of this century, China has launched the revitalization of the Northeast and made adjustments at any time according to the requirements of the times, with the aim of solving the development problems including urban shrinkage and promoting the coordinated development of regional economy [29].

### 4.2. The Spatio-Temporal Changes of Urban Vitality in the Three Northeastern Provinces of China Are Obvious

The concept of urban vitality has always had practical significance [48]. In recent years, urban vitality has attracted more and more attention and discussion, but the related research results are not fruitful enough. From 2010 to 2018, the urban vitality of most cities in the three northeastern provinces of China has improved slightly, but the relative size has not changed much. From the overall performance of the nine years, the urban vitality of Dalian, Shenyang, Changchun and Harbin has been at a high level for a long time, which matches their functional positioning in this region. It is found that with the change of time, the urban vitality values of the three northeastern provinces of China tend to gather in space. Dalian, Shenyang, Changchun and Harbin have formed their own centers, which promote the vitality of surrounding cities with their own high values of urban vitality through the functions of factor flow and external radiation, and then form larger clusters of medium and high values and even become corridors, such as the Liaodong Peninsula city group. In contrast, cities with low urban vitality will also gather together, such as the cities near northern border of Heilongjiang Province. Further, when the gathering phenomenon caused by different urban vitality evolves to a certain stage, there may be serious polarization phenomenon. This process is driven by the flow of elements within the region, although it can enhance the local urban vitality to a certain extent, it will eventually lead to the imbalance or even polarization of urban development, which is not conducive to regional integration and coordinated urban development.

This research realizes the concretization of abstract indicators through the visualization of data, which makes the law of urban vitality in time and space in the three northeastern provinces of China very obvious. In addition, it is found that although the spatial pattern of urban vitality in the three northeastern provinces of China is prominent, the numerical change law is extremely weak. This situation may be mainly caused by two reasons: first, the development dilemma of cities in the three northeastern provinces of China leads to the slow improvement of urban vitality; second, the statistical data used in this research include stock and incremental indicators, which have different target orientations in depicting urban vitality. Stock indicators’ absolute values gradually increase with time while incremental indicators’ relative values change dynamically in regional association. There are certain contradictions and disputes, which should be paid attention to in the follow-up research. In terms of measures, China’s central and local governments have also made efforts to curb the polarization and differentiation of urban vitality through the revitalization of the Northeast and related policies, and to solve the core problems existing in this area with the goal as the guide [29].

### 4.3. Urban Shrinkage Is Significantly Related to Urban Vitality

From 2010 to 2018, there is a significant positive correlation between urban shrinkage scores and comprehensive scores of urban vitality in the three northeastern provinces of China. Because the urban shrinkage scores are negative and the comprehensive scores of urban vitality are positive, we think that in general, the lower the degree of urban shrinkage is, the higher the urban vitality is. It is found that, in the research period, the urban shrinkage is related to the urban vitality every year, and the correlation reaches the maximum value in the middle year (2014). This paper holds that, on the one hand, 2014 is included in the years of most cities’ shrinkage process, and on the other hand, 2014 is considered as the beginning period of “New Northeastern Phenomenon” [29]. In this case, the relationship between urban shrinkage and urban vitality is typical, which reflects the intermediate state of cities in the three northeastern provinces of China after the new development dilemma. Urban shrinkage in the three northeastern provinces of China began with population reduction. Next, population reduction will bring a chain of reactions in many aspects, such as economy and society, and then each city will enter into different shrinkage phases, which will lead to functional transformations itself. These changes can further affect the population and cause the flow of other different elements in the region. However, the shrinkage degree of each city is different, and the results of urban shrinkage in different stages on the economic strength, social structure, cultural characteristics, ecological environment and space shaping of each city are naturally different. This paper posits that though the relationship between urban shrinkage and urban vitality shows that the former has a significant impact on the latter, urban shrinkage is not the decisive factor of urban vitality, nor should it be an obstacle to limit urban vitality. Urban shrinkage, as an important process of urbanization, is also an expression of urban state, which is caused by various reasons and also has multiple influences. Urban vitality, also as a manifestation of urban state, aims to pay attention to the quality of the city [49]. Studying the correlation between urban shrinkage and urban vitality makes up for theoretical shortcomings in related fields, explores the development needs of shrinking cities at current stage, and provides theoretical guidance for the planning and design of shrinking cities. Combined with the viewpoints mentioned above and the existing research, China should reasonably guide the urban development such as “smart shrinkage” by compiling the stock and reduction plannings, and effectively enhance the vitality of cities with the help of the potential energy of urban shrinkage rather than the pressure of it [16,24].

### 4.4. Urban Shrinkage Has Different Impacts on All Dimensions of Urban Vitality

There are obvious differences in the influence degree of urban shrinkage on urban economic vitality, social vitality, cultural vitality, environmental vitality and spatial vitality in the three northeastern provinces of China, which also reflects to some extent that urban vitality is composed of many aspects of vitality, and there are many influencing factors, but the effects of different factors are different [50]. According to the five groups of correlation coefficient values, urban shrinkage has the greatest impact on economic vitality. This is because, whether it is judged from the actual situation or from the index system, the level of urban economy is an important symbol of each stage of urban development, and it is also the main representation of urban vitality on the macro scale [40]. Besides population reduction, the result of urban shrinkage is economic slowdown or even recession, and the “Northeast Phenomenon” at the end of last century and the “New Northeast Phenomenon” in this century mainly explain the economic phenomena. The impact of urban shrinkage on cultural vitality and environmental vitality comes second. Cultural vitality and environmental vitality, as factors closely related to people, respectively reflect the level of urban carrying capacity and soft power. The changes of both are accompanied by urban shrinkage, and they can often reflect the key of cities’ energy beyond the population and economic level [49,51]. The impact of urban shrinkage on social vitality and spatial vitality is certainly small. This is because although the evaluation indicators of social vitality all revolve around human activities, they often have a certain quantitative basis and local characteristics, and the data indicators are mostly the ratios of dimension removal, so it is difficult to change obviously in a limited time due to urban shrinkage. Spatial vitality often reflects the level of urban public facilities and urban construction. Under the background of continuous optimization of territory spatial planning system and urban infrastructure construction level in China, spatial vitality will generally not decrease or stagnate with the shrinkage of cities [52,53]. This paper analyzes the influence mechanism of urban shrinkage on various dimensions of urban vitality, hoping to provide enlightenment for future related research and urban planning practice.

### 4.5. Limitations and Future Research

At present, the academic research on the effect of urban shrinkage is still insufficient, especially the urban vitality directly related to urban competitiveness. According to the “two-step diagnostic method”, this paper identifies the urban shrinkage, measures the urban vitality with the help of macro indicators, and establishes a correlation model, which to some extent fills the gap in the research on the correlation between urban shrinkage and urban vitality, and contributes to the research of urban shrinkage effect. However, due to the limitations of our thinking, the research also shows the following deficiencies:

(1) Urban shrinkage and urban vitality are both complex systems, and their respective influencing factors are numerous. This paper comprehensively constructs the corresponding evaluation system through the economic and social aspects of the existing research, but the evaluation criterions of the two in the existing research are still inconclusive. In the future, it is still necessary to continuously improve and supplement and add a reasonable index system. On the one hand, it is suggested that identify the urban shrinkage in more detail, grasp its essential characteristics, and analyze the shrinkage effect in combination with the integrated development of urban and rural areas in China [54]. On the other hand, the increment index and stock index of urban vitality should be judged separately, and a more accurate trend of urban vitality will be obtained.

(2) This research explores the relationship between urban shrinkage and urban vitality and confirms that there is a significant correlation between them, which has produced different effects in economic and social dimensions. However, the correlation between the two is not only reflected in the change of correlation coefficient and simple line chart. In the future, it is necessary to deeply consider the spatial heterogeneity of urban shrinkage and urban vitality, and explore their spatial and temporal changes, which can be further analyzed by, for example, Geographic Weighted Regression Model (GWR) or Geographic and Time Weighted Regression Model (GTWR).

(3) In terms of data, the statistical yearbook is selected in this paper. Although the source is reliable, the overall data is a little rough. Though some missing data can be supplemented by interpolation and other methods, it will affect the overall accuracy more or less. In the future, extracting the urban built-up areas from the data pretreated by radiometric calibration or National Polar—orbiting Partnership/Visible Infrared Imaging Radiometer Suite nighttime light data, we can obtain more accurate research areas than the municipal districts in statistical data, and then carry out the research on county-level cities to be closer to the current situations of China [55]. Using web crawler and other technical means to collect big data, POI, industry data, etc. of the cities at the street and community level, we can understand the urban vitality at the micro scale and make a more detailed measurement, which meets the requirements of sustainable development and puts forward new suggestions for urban development [20,56,57].

(4) At present, the research only focuses on the three northeastern provinces of China. However, the phenomenon of urban shrinkage is becoming more and more obvious, and similar phenomena are gradually appearing in cities in China’s Yangtze River Economic Belt [58]. In the future research, we will try to explore the relationship between urban shrinkage and urban vitality in other regions, such as other economic sectors and resource-based cities in China, etc. In addition, we will consider the impact of the COVID-19 epidemic on urban development and draw a more comprehensive and objective conclusion.

## 5. Conclusions

Urban shrinkage and urban vitality are unavoidable topics in the process of current urbanization, and it is important to study the relationship between them for the future urban development. In this study, 34 cities in the three northeastern provinces of China are selected as the research objects, and the statistical data from 2010 to 2018 are used. First, the “two-step diagnostic method” is used to identify the urban shrinkage, and then EWM is used to calculate the annual urban vitality of each city. Finally, the correlation between urban shrinkage and urban vitality is analyzed. The findings of this research are as follows: (1) 61.76% of cities in the three northeastern provinces of China have experienced urban shrinkage. Urban shrinkage has become a regional phenomenon in the three northeastern provinces of China, and it is also an inevitable process in China and even other parts of the world. (2) From 2010 to 2018, the urban vitality of the three northeastern provinces of China was generally steady and rising, and the comprehensive scores of urban vitality show a trend of spatial agglomeration according to the height. Yearly, the urban vitality values of Municipalities with Independent Planning Status under the National Social and Economic Development (Dalian) and provincial capitals have been at a high level for a long time, reflecting the trend of factor flow and the imbalance of internal development of the region. (3) There is a significant positive correlation between urban shrinkage and urban vitality in the three northeastern provinces of China. The score of urban shrinkage is negative, and the value of urban vitality is positive, so in general, the lower the degree of urban shrinkage is, the higher the urban vitality is. Urban shrinkage affects the vitality of cities by changing the development factors, and then establishes a correlation to it. (4) The influence of urban shrinkage on each dimension of urban vitality is different according to the different results of correlation coefficient. The correlation between urban shrinkage and economic vitality is the most significant and stable, indicating that the economic level is still an important symbol of urban development; The correlation between urban shrinkage and cultural vitality and environmental vitality is at a medium level, which indicates that the cultural characteristics and environmental carrying capacity cities will be affected by urban shrinkage, but it can also reflect the essence of vitality and get rid of external effects appropriately. The correlation between urban shrinkage and social vitality and spatial vitality is not significant, which indicates that the local differences of urban social structure are obvious, and the sustainable optimization of urban spatial structure in China is difficult to change due to urban shrinkage. The relationships between urban shrinkage and each dimension of urban vitality are different, which affects the level of correlation.

The “two-step diagnostic method” used in this research innovatively gives a system of multi-factor identification of shrinking cities on the basis of fully considering China’s national conditions, which provides a new idea for the research of urban shrinkage identification. At the same time, the significant correlation between urban shrinkage and urban vitality found in this research explains some effects of urban shrinkage to a certain extent, and also reveals the theoretical scientificity of the interactions between urban shrinkage and urban vitality, which can provide a theoretical basis for the corresponding changes in the process of urbanization, and then better guide the high-quality development of cities in the future. Based on this, the three northeastern provinces of China should adhere to strategic determination, and all parts of the world should reach a consensus, aiming at achieving high-quality regional and urban development, reasonably grasping the relationship between urban shrinkage and urban vitality, and guiding their benign interactions.

## Figures and Tables

**Figure 1 ijerph-19-10650-f001:**
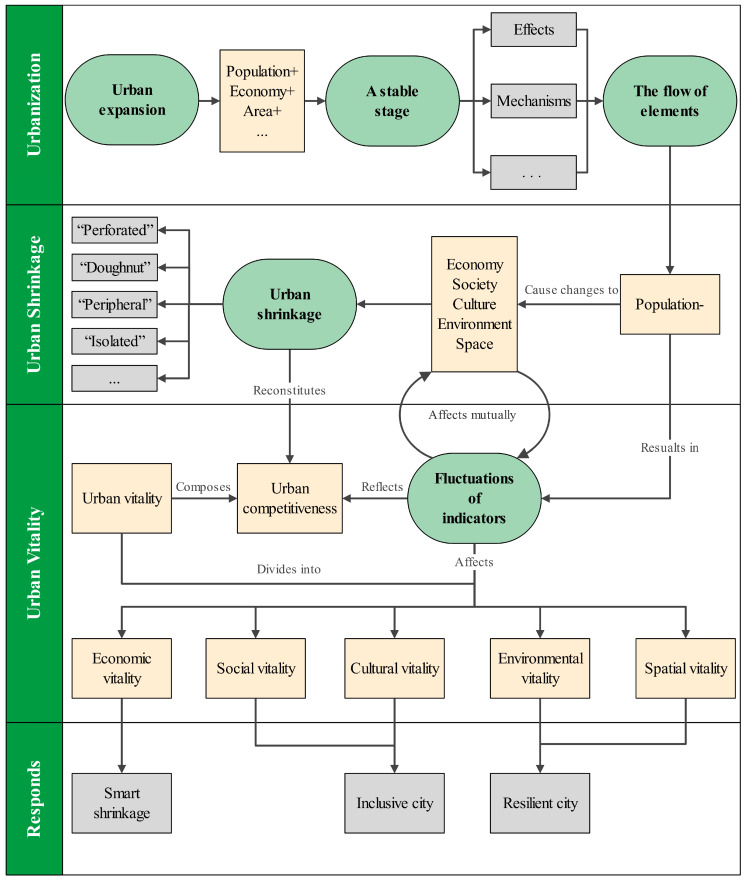
The theoretical connection between urban shrinkage and urban vitality.

**Figure 2 ijerph-19-10650-f002:**
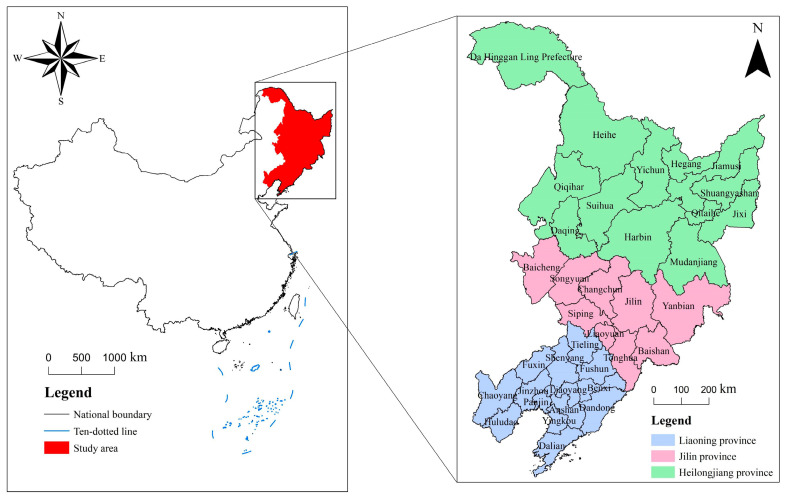
Geographical location of the three northeastern provinces of China. Note: The figure number is GS(2019)1822.

**Figure 3 ijerph-19-10650-f003:**
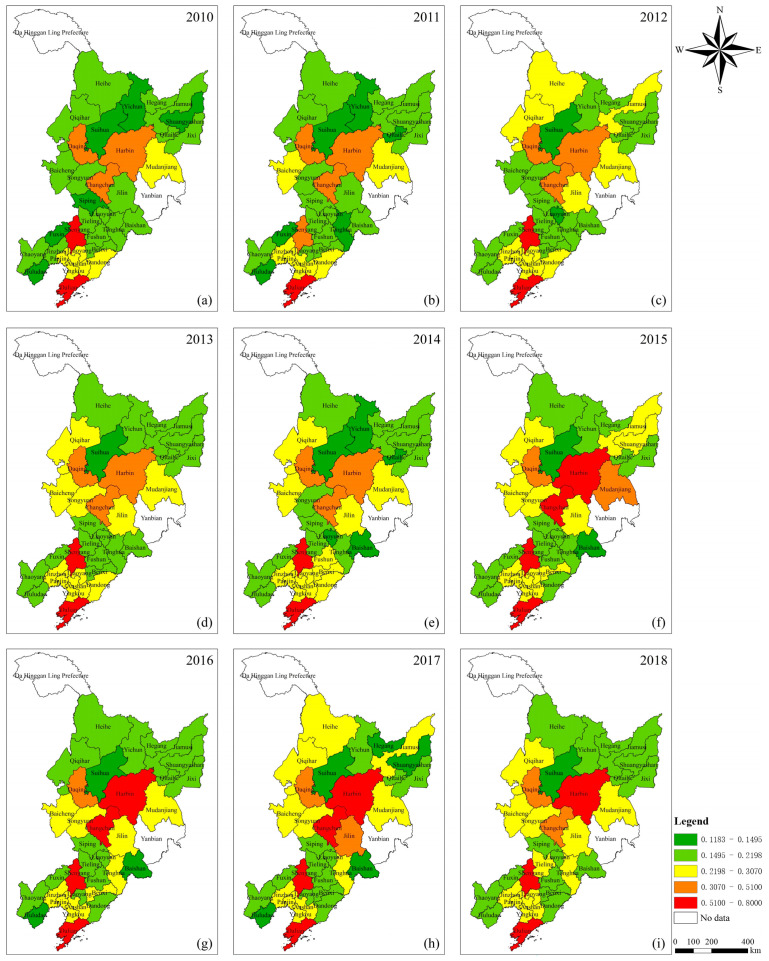
Variations of urban vitality of cities in three northeastern provinces of China from 2010 to 2018. Note: In order to visualize the results better, the municipal districts are shown as administrative regions. And (**a**–**i**) respectively correspond to the subfigures of 2010–2018, marking their correct order.

**Table 1 ijerph-19-10650-t001:** Index system of urban shrinkage.

Object	Demographic Dimension	Economic Dimension	Social Dimension
Index	I1: The annual average population of municipal districts	I2: Gross domestic product (GDP)	I5: Total retail sales of social consumer goods
		I3: Per capita disposable income	I6: The number of doctors
		I4: The proportion of tertiary industry output value to GDP	

**Table 2 ijerph-19-10650-t002:** Stages of urban shrinkage and the classification criteria.

Stage of Urban Shrinkage	Criterions *
Non-shrinkage	X_1_ > 0
Slight shrinkage	−3 ≤ X_1_ < 0
Preliminary shrinkage	−5 ≤ X_1_ < −3 and A ≥ −20
Middling shrinkage	−10 ≤ X_1_ < −5, or X_1_ < −10 and A > 0, or −5 ≤ X_1_ < −3 and A ≤ −20
Severe shrinkage	X_1_ ≤ −10 and A < 0

* The criterions in the table are compiled according to the existing research [28,33].

**Table 3 ijerph-19-10650-t003:** Index system of urban vitality.

Dimension	Indicator	Explain of the Index	Number of the Index	References
Economic vitality	Income level	Per capita disposable income	1	[40,42]
	Residents’ Consumption level	Total retail sales of social consumer goods	2	[42,43]
	Investment of foreign capital	The actual amount of foreign capital used	3	[40,43]
	Economic growth	Growth rate of GDP	4	[40]
	Industrial structure	The proportion of tertiary industry output value to GDP	5	[40,43]
	Market size	The number of enterprises in domestic-funded industries above the scale	6	[40]
Social vitality	Innovation capacity	The number of annual patent applications per 10,000 people	7	[40,44]
	Unemployment	Percentage of registered unemployed people in cities	8	[44]
	Demographic Changes	Natural population growth rate	9	[38]
	Population density	The number of people per unit area	10	[21,44]
	Medical insurance	The number of doctors per 10,000 people	11	[38]
Cultural vitality	Cultural development	The number of books in public libraries per 100 people	12	[38]
	cultural consumption	The proportion of employees in education, culture, sports and entertainment	13	[38,45]
Environmental vitality	Greening level	Per capita park green space area	14	[40,45]
	Water supplying capacity	Per capita water supply	15	[40,43]
	Power supplying capacity	Per capita electricity consumption	16	[40,43]
	Discharging capacity	The density of drainage pipe	17	[45]
Spatial vitality	Accessibility of traffic	The density of roads in cities	18	[20,38]
	Efficiency of public transportation	The number of buses per 10,000 people	19	[40,43]
	The level of living space	Area of per capita residential land for construction	20	[38,40]

**Table 4 ijerph-19-10650-t004:** Identification results of shrinking cities in three northeastern provinces of China.

City	The Starting Year of Shrinking	The End Year of Shrinking	Population Growth Rate (%)	The First Step of Judgment	The Second Step of Judgment	Stage of Shrinking
Harbin			0	No pass		Non-shrinkage
Qiqihar	2010	2018	−6.6767	Pass	−19.7208	Middling shrinkage
Jixi	2010	2018	−12.9615	Pass	−3.2914	Severe shrinkage
Hegang	2010	2018	−11.3279	Pass	No pass	Middling shrinkage
Shuangyashan	2012	2014	−6.0070	Pass	−5.5404	Middling shrinkage
	2016	2018				
Daqing	2013	2018	0	No pass		Non-shrinkage
Yichun	2012	2017	−7.5342	Pass	−3.8713	Middling shrinkage
Jiamusi	2012	2017	−5.5844	Pass	No pass	Middling shrinkage
Qitaihe			−18.9583	Pass	−10.9921	Severe shrinkage
Mudanjiang			0	No pass		Non-shrinkage
Heihe	2011	2017	0	No pass		Non-shrinkage
Suihua			−9.2683	Pass	−4.0642	Middling shrinkage
Changchun			0	No pass		Non-shrinkage
Jilin	2011	2015	−0.7807	Pass	No pass	Slight shrinkage
Siping	2011	2016	−4.7806	Pass	−8.6928	Preliminary shrinkage
Liaoyuan			0	No pass		Non-shrinkage
Tonghua	2010	2013	−1.9929	Pass	−11.0414	Slight shrinkage
	2015	2018				
Baishan	2011	2018	−12.0755	Pass	No pass	Middling shrinkage
Songyuan			0	No pass		Non-shrinkage
Baicheng	2012	2016	−3.6735	Pass	−1.7364	Preliminary shrinkage
Shenyang			0	No pass		Non-shrinkage
Dalian			0	No pass		non-shrinkage
Anshan	2013	2018	−2.5000	Pass	−11.0960	Slight shrinkage
Fushun	2013	2018	−4.8905	Pass	−6.7867	Preliminary shrinkage
Benxi	2010	2018	−5.9222	Pass	−11.0951	Middling shrinkage
Dandong	2011	2016	−1.0256	Pass	No pass	Slight shrinkage
Jinzhou			0	No pass		Non-shrinkage
Yingkou			0	No pass		Non-shrinkage
Fuxin	2013	2016	−2.2368	Pass	−9.4948	Preliminary shrinkage
Liaoyang	2014	2017	−1.9767	Pass	−0.9507	Slight shrinkage
Panjin			0	No pass		Non-shrinkage
Tieling	2011	2016	−3.7209	Pass	−22.2522	Middling shrinkage

**Table 5 ijerph-19-10650-t005:** Scores and ranks of urban vitality in three northeastern provinces of China.

City	2010		2011		2012		2013		2014	
	Score	Rank	Score	Rank	Score	Rank	Score	Rank	Score	Rank
Harbin	0.37	4	0.50	3	0.44	3	0.42	4	0.51	3
Qiqihar	0.20	16	0.21	17	0.25	11	0.23	15	0.24	15
Jixi	0.17	24	0.18	22	0.19	26	0.17	30	0.18	24
Hegang	0.19	18	0.18	20	0.22	17	0.20	25	0.17	29
Shuangyashan	0.15	29	0.16	26	0.21	20	0.21	18	0.17	27
Daqing	0.35	5	0.33	5	0.37	5	0.35	5	0.40	5
Yichun	0.14	32	0.15	30	0.19	27	0.17	31	0.15	30
Jiamusi	0.18	20	0.20	18	0.23	15	0.21	20	0.20	20
Qitaihe	0.19	17	0.14	33	0.17	29	0.18	27	0.15	31
Mudanjiang	0.26	7	0.25	6	0.28	6	0.28	7	0.29	8
Heihe	0.18	22	0.21	16	0.23	16	0.22	17	0.22	19
Suihua	0.14	33	0.14	31	0.14	34	0.13	34	0.12	34
Changchun	0.37	3	0.36	4	0.41	4	0.43	3	0.43	4
Jilin	0.22	12	0.22	13	0.24	13	0.25	14	0.26	12
Siping	0.14	31	0.16	25	0.18	28	0.18	26	0.18	25
Liaoyuan	0.19	19	0.15	28	0.15	33	0.16	33	0.14	32
Tonghua	0.15	28	0.15	29	0.17	30	0.21	22	0.18	26
Baishan	0.16	27	0.17	24	0.17	31	0.16	32	0.13	33
Songyuan	0.17	26	0.19	19	0.21	23	0.25	13	0.22	18
Baicheng	0.20	14	0.23	10	0.21	19	0.26	11	0.24	13
Shenyang	0.56	2	0.51	2	0.58	2	0.57	2	0.61	2
Dalian	0.68	1	0.58	1	0.67	1	0.68	1	0.71	1
Anshan	0.27	6	0.25	7	0.27	9	0.27	8	0.31	7
Fushun	0.18	21	0.18	21	0.21	22	0.21	23	0.23	17
Benxi	0.18	23	0.17	23	0.21	21	0.21	19	0.24	16
Dandong	0.22	11	0.24	9	0.28	8	0.26	10	0.26	11
Jinzhou	0.20	15	0.23	12	0.26	10	0.26	9	0.27	9
Yingkou	0.25	8	0.24	8	0.28	7	0.29	6	0.31	6
Fuxin	0.14	30	0.14	32	0.20	24	0.17	28	0.20	21
Liaoyang	0.22	10	0.22	14	0.23	14	0.22	16	0.24	14
Panjin	0.24	9	0.23	11	0.25	12	0.26	12	0.27	10
Tieling	0.21	13	0.21	15	0.22	18	0.21	21	0.19	22
Chaoyang	0.17	25	0.15	27	0.20	25	0.20	24	0.18	23
Huludao	0.12	34	0.14	34	0.16	32	0.17	29	0.17	28
**City**	**2015**		**2016**			**2017**			**2018**	
	**Score**	**Rank**	**Score**	**Rank**		**Score**		**Rank**	**Score**	**Rank**
Harbin	0.54	4	0.55	4		0.60		3	0.67	2
Qiqihar	0.25	13	0.21	18		0.23		14	0.24	14
Jixi	0.18	27	0.18	25		0.20		19	0.18	27
Hegang	0.17	29	0.16	30		0.15		30	0.16	32
Shuangyashan	0.23	17	0.18	23		0.14		32	0.16	31
Daqing	0.39	5	0.38	5		0.39		5	0.38	5
Yichun	0.18	26	0.17	28		0.18		25	0.18	29
Jiamusi	0.22	18	0.19	20		0.22		17	0.18	28
Qitaihe	0.19	25	0.16	31		0.17		27	0.18	26
Mudanjiang	0.33	6	0.28	7		0.23		13	0.29	7
Heihe	0.21	22	0.18	24		0.23		15	0.20	20
Suihua	0.12	34	0.11	34		0.12		33	0.13	34
Changchun	0.57	3	0.62	1		0.67		2	0.50	4
Jilin	0.28	9	0.29	6		0.32		6	0.27	10
Siping	0.17	30	0.18	26		0.22		18	0.19	21
Liaoyuan	0.16	32	0.17	29		0.18		26	0.19	24
Tonghua	0.19	24	0.22	15		0.24		11	0.23	16
Baishan	0.15	33	0.13	33		0.15		31	0.15	33
Songyuan	0.26	12	0.24	12		0.28		7	0.27	8
Baicheng	0.29	8	0.27	8		0.28		8	0.26	12
Shenyang	0.58	2	0.56	3		0.57		4	0.59	3
Dalian	0.59	1	0.58	2		0.67		1	0.72	1
Anshan	0.30	7	0.26	10		0.26		9	0.29	6
Fushun	0.22	20	0.21	17		0.20		21	0.21	18
Benxi	0.25	15	0.22	14		0.17		28	0.19	22
Dandong	0.22	19	0.19	22		0.19		23	0.20	19
Jinzhou	0.27	10	0.26	9		0.23		16	0.27	11
Yingkou	0.27	11	0.23	13		0.25		10	0.27	9
Fuxin	0.20	23	0.19	21		0.18		24	0.18	25
Liaoyang	0.23	16	0.22	16		0.20		20	0.23	15
Panjin	0.25	14	0.25	11		0.23		12	0.26	13
Tieling	0.21	21	0.20	19		0.20		22	0.22	17
Chaoyang	0.17	28	0.17	27		0.16		29	0.19	23
Huludao	0.16	31	0.14	32		0.10		34	0.16	30

**Table 6 ijerph-19-10650-t006:** The correlation coefficient between urban shrinkage and urban vitality.

Correlation Coefficient	2010	2011	2012	2013	2014	2015	2016	2017	2018
Urban vitality	0.386	0.451	0.438	0.469	0.488	0.457	0.463	0.437	0.457
Economic vitality	0.363	0.401	0.386	0.402	0.411	0.398	0.426	0.408	0.401
Social vitality	0.123	0.182	0.165	0.262	0.326	0.235	0.344	0.276	0.249
Cultural vitality	0.467	0.446	0.485	0.446	0.449	0.403	0.360	0.321	0.269
Environmental vitality	0.291	0.371	0.305	0.307	0.562	0.437	0.369	0.434	0.609
Spatial vitality	0.172	0.164	0.113	0.154	0.088	−0.029	0.01	−0.068	0.149

## Data Availability

No new data were created or analyzed in this study. Data sharing is not applicable to this article.

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
