# Peer review of "Urban Shrinkage and Urban Vitality Correlation Research in the Three Northeastern Provinces of China"

_ijerph, 2022, doi:10.3390/ijerph191710650_

Round 1

Reviewer 1 Report

With the continuous advancement of globalization and urbanization, the phenomenon of urban shrinkage has gradually evolved into a worldwide social and economic development problem, constantly emerging in all corners of the world, which has also aroused concern in the academia. Many theoretical and empirical analyses have been carried out around the concept and connotation of urban shrinkage together with its identification and evaluation systems, spatio-temporal evolution characteristics and types, formation mechanism and response path, and the research results have been very fruitful. However, there are regional heterogeneity in the background and mechanism of urban shrinkage, and it is the consensus of current researches to explore the common law of urban shrinkage based on "the experience of each country". Secondly, the research contents focused on the aspects of identification-pattern-mechanism-response, and the researches on urban shrinkage effect is relatively few, especially the researches on the relationship between urban shrinkage and urban vitality. Based on this situation, the paper selected the three northeastern provinces of China with the most typical urban shrinkage phenomenon to make an empirical analysis on the correlation between urban shrinkage and urban vitality, and accordingly put forward the “two-step diagnostic method” for shrinking cities in Chinese context, which was of novelty and practical significance. The logic of the paper was clear, the structure was reasonable, and the research conclusion was persuasive. It is a nice empirical analysis paper in general, and it is recommended to accept publication. However, after reading the paper, I have comments and suggestions to improve as follows.

1.     Try to improve the expression of introduction section to organize your ideas of urban shrinkage and urban vitality better with a figure.

2.     Why did you choose the “two-step diagnostic method” in line 226? What typical advantages does the method have?

3.     What’s the possible reason of your choice of selecting the natural breakpoints of the comprehensive scores of 2014 in line 363-364? You’d better make an explanation.

4.     The highlights which are innovative enough needs to be strengthened felicitously and discussed valuably in line 566-586 or line 670-679.

5.     It seemed that the summary of the characteristics of each dimensions of urban vitality was missing in conclusion section.

6.     Check the conjunctions and transitional sentences to make the paper’s expressions smoother.

Reviewer 2 Report

The study examines the urban shrinkage and urban vitality connection in China. While the study offers some insights, there are some key aspects to improve the quality of the paper.

Abstract has some typos. (i.e., isn’t!)

Abstract indicators population/economy/society are not clear enough and population may go under society category

Intro: The transition from shrinkage to vitality is too abrupt. It needs a smooth one

Materials and Methods: The selection criteria of 34 cities is not convincing enough. It should be expanded and highlighted in the context of urban shrinkage and urban vitality.

Are the data publicly available? If so, the author(s) should provide supplemantary materials or external source links.

How did the author(s) decide the table 1 criteria? Urban shrinkage is a complex phenomenon and it needs a great justification here.

Same thing applied fort he table 3 too for urban vitality criteria.

Is table 2 coming from the reference in the paper of the author(s) created based on that?

 Legends of figures need revising

Where is the correlation table? Not the figure one

Conclusion needs a great expansion based on such efforts.

Reviewer 3 Report

Thank you for giving me this opportunity to read the manuscript entitled "Urban Shrinkage and Urban Vitality Correlation Research in the Three Northeastern Provinces of China". The topic of this manuscript is interesting and would be a good contribution to this field. I think it could be considered for publication in International Journal of Environmental Research and Public Health once the following issues are addressed.

1.     Please replace the keywords that already appear in the manuscript's title with close synonyms or other keywords, which will also facilitate your paper to be searched by potential readers.

2.     The maps in Fig. 1 and Fig. 2 should be improved in terms of resolution. Besides, test information in these Figs is too small and can not be read clearly.

3.     It is good to see the authors try to make the analysis based on datasets with annual update frequency, but the uncertainties caused by how the authors addressed the missing data should be discussed.

4.     The spatial variation in relationships between urban shrinkage and vitality due to spatial heterogeneity needs to be discussed. The authors do not necessarily use geographically weighted regression (GWR) models or geographically and temporally weighted regression (GTWR) models to reconduct the analysis. However, spatial heterogeneity is still recommended to be appropriately considered in this manuscript.

5.     Only one compass, one scale, and one legend will be enough in Fig.2.

6.     Line 119-121: Some newly published papers regarding the using urban vitality information to address public health and urban environment issues could be added as references to support the statement here. For example, the papers titled "Dynamic assessments of population exposure to urban greenspace using multi-source big data".

7.     "Limitation" should be added as a sub-section of "Discussion".

8.      Some grammatical errors exist in the manuscript. Therefore, a critical review of the manuscript language will improve readability.

Round 2

Reviewer 2 Report

Thank you for addressing all my comments

Reviewer 3 Report

Thank you for giving me this opportunity to read the revised version of the manuscript titled "Urban Shrinkage and Urban Vitality Correlation Research in the Three Northeastern Provinces of China", and for the detailed responses to my earlier comments. I am satisfied with this revised version, and I think it is acceptable now.